# Impact of Surgical Care Bundle on Surgical Site Infection after Non-Reconstructive Breast Cancer Surgery: A Single-Centre Retrospective Comparative Cohort Study

**DOI:** 10.3390/cancers15030919

**Published:** 2023-02-01

**Authors:** Kian Chin, Fredrik Wärnberg, Anikó Kovacs, Roger Olofsson Bagge

**Affiliations:** 1Department of Surgery, Sahlgrenska University Hospital, 41345 Gothenburg, Sweden; 2Institute of Clinical Sciences, Sahlgrenska Academy, University of Gothenburg, 41345 Gothenburg, Sweden; 3Department of Clinical Pathology, Sahlgrenska University Hospital, 41345 Gothenburg, Sweden; 4Wallenberg Center of Translational Medicine, University of Gothenburg, 41345 Gothenburg, Sweden

**Keywords:** breast cancer, surgical site infection, care bundle, neoadjuvant chemotherapy, mastectomy, breast conservation, axillary lymph node clearance, sentinel lymph node biopsy

## Abstract

**Simple Summary:**

Wound infection or so-called surgical-site infection (SSI) is a common occurrence after surgery. For some breast cancer patients, these infections can lead to delays in starting their onward systemic treatments such as chemotherapy and/or radiotherapy after surgery. More importantly, treatment delays in patients can lead to worsened overall survival. SSIs also have considerable negative impacts on financial and staffing resources in healthcare. The World Health Organization has recommended the usage of a surgical care bundle (SCB), which is a group of preventative measures that are effective in reducing SSIs. However, the impact of care bundles on SSIs has not been well documented in the context of breast cancer surgery. Therefore, we aimed to investigate the outcomes of SSI following the implementation of a surgical care bundle protocol for non-reconstructive breast cancer surgery.

**Abstract:**

Background: Surgical-site infections (SSIs) are the commonest cause of healthcare-related infections. Although a surgical care bundle (SCB), defined as a group of preventative measures, is effective in reducing SSIs, it has not been well documented in breast cancer surgery. We aimed to investigate the impact of SCB on SSI. Methods: A single-centre retrospective comparative cohort study between 2016 and 2020 was carried out. An SCB including eight different measures was implemented in October 2018 at Sahlgrenska University Hospital, Sweden. Patients who underwent non-reconstructive breast cancer surgery were included for analysis. The primary endpoint was SSI within 30 days after surgery. Results: Overall, 10.4% of patients (100/958) developed SSI. After SCB implementation, the overall SSI rate reduced from 11.8% to 8.9% (*p* = 0.15). The largest SSI rate reduction was seen in the subgroup that underwent breast conservation and sentinel lymph node biopsy (SLNB), from 18.8% to 9.8% (*p* = 0.01). In this multivariable analysis adjusting for patient and treatment factors, the implementation of SCB resulted in a statistically significant reduction in SSI risk (OR 0.63, 95% CI 0.40–0.99, *p* = 0.04). Conclusions: The implementation of a SCB could reduce the incidence of SSI in breast cancer surgery.

## 1. Introduction

Breast cancer is one of the most common cancers and the leading cause of death amongst females. Surgery is an important and effective treatment, either alone or together with endocrine treatment, chemotherapy, targeted treatments and radiotherapy. However, surgical site infection (SSI) is also a common complication in the immediate postoperative period, which can have negative impacts on patient safety, hospital resources and aesthetic surgical outcome of the breast. Postoperative SSI can also lead to delays in adjuvant treatments, which is associated with an increased risk of breast cancer recurrence [1,2].

Between 2013 and 2018, 34% of healthcare-related patient injuries in Sweden were due to infections, of which up to a third were classified as SSI. The reported overall SSI rate was 1% to 3% but there were no specific data relating to breast surgery [3]. The average national cost of all healthcare-related injuries was estimated to be up to USD 220 million annually [3]. Overall, worldwide incidence of SSI after breast surgery has been reported to vary between 2% and 30% [4,5,6]. Due to inherent variations in methods of data collection, it is difficult to extrapolate published data for useful comparisons. Therefore, in the absence of generally reliable and comparable SSI data, it is paramount for individual breast cancer units to establish their own data to drive infection prevention programs and maximise patient safety.

There are multiple factors associated with increased risk of wound healing problems and SSI after breast surgery, including smoking, body mass index, diabetes and hypothermia [7,8,9]. Recent developments in oncoplastic techniques have also meant more complex surgery with increased postoperative morbidities such as SSI [10]. In addition, there are reports that neoadjuvant chemotherapy (NACT) may lead to increased risk of SSI, whereas other retrospective studies reported similar overall short-term surgical site morbidity irrespective of NACT [11,12,13]. Since the usage of NACT has become more commonly used to treat certain breast cancer subtypes, research in preventative measures that can lower SSI risks has become more clinically pertinent.

A surgical care bundle (SCB) is defined as a group of strategies that can be implemented as part of pre-, intra-, or perioperative care routines to minimize SSI amongst patients undergoing surgery [14]. The World Health Organization (WHO) issued global guidelines for the prevention of SSI and the relevant effective care bundle components [15]. It was recommended that a SCB should be constructed using five to six action points derived from good evidence [14]. In breast-cancer-related reconstruction surgery, SSI had been reported in up to 30% of patients [16], and SSI could possibly be reduced with the adoption of a care bundle protocol [17]. Similarly, the effectiveness of SCB in reducing SSI after colorectal surgery was up to 7% in some studies [18,19].

However, the impact of the care bundle has not been well determined in non-reconstructive breast cancer surgery. Therefore, in this study, the primary aim was to describe SSI and investigate the impact of implementing SCB on SSI amongst patients who had non-reconstructive breast conservation surgery (BCS) and mastectomy. Secondary aims included the adherence rate of SCB after implementation and potential adverse effects of SCB with reference to rates of thromboembolic events (pulmonary emboli and deep vein thrombosis), seroma aspiration, day surgery, re-operation and delayed start of adjuvant chemoradiotherapy treatment.

## 2. Materials and Methods

### 2.1. Surgical Care Bundle

In 2018, the WHO introduced thirty recommendations for preventing SSI, consisting of ten preoperative, fifteen intraoperative and five postoperative measures [15]. In this study, an SCB was introduced with five recommendations from the WHO including (1) preoperative wash using soap and water instead of chlorhexidine, (2) prophylactic antibiotics (intravenous 2 g cloxacillin or 600 mg clindamycin) to be given from 2 h up to 30 min before skin incision, (3) wound irrigation with normal saline, (4) use of monofilament triclosan coated sutures and (5) wound dressing with surgical tapes, only avoiding the need for regular changes of dressing, which can itself be resource-intensive and a potential risk factor for SSI. Other incorporated non-WHO measures were (6) routine local anaesthetic infiltration to minimise pain in order to facilitate ambulatory surgery whenever possible and (7) avoiding routine use of low molecular weight heparin (LMWH) as thromboprophylaxis since postoperative haematoma was considered a risk factor for SSI. The routine use of (8) surgical drains was considered a low-evidence-based practice that hindered day-case surgery and was therefore not recommended.

### 2.2. Study Aim, Design, and Outcome Measures

The primary aim was to investigate the impact of SCB in reducing SSI using a multivariate analysis, adjusting for patient and tumour characteristics. This was a retrospective cohort study of patients who underwent non-reconstructive-related breast cancer surgery between January 2016 and December 2020 at Sahlgrenska University Hospital, Sweden. The SCB was implemented in October 2018. The period between November 2018 and January 2019 was considered an early SCB introductory phase and therefore excluded from the study. Patients with SSIs were identified through investigation of the electronic medical records (Melior) as well as by searching the Cognos Analytics^TM^ hospital database using infection International Classification of Diseases (ICD) code T81.4. 

The primary outcome measurement was 30-day postoperative SSI adjusted for age, BMI, smoking, diabetes, types of surgery, NACT and seroma aspirations. Surgical site infection was diagnosed according to the Centre for Disease Control and Prevention criteria that include presence of erythema, localized swelling, pain, purulent discharge with or without fever, or positive bacterial culture, as well as diagnosis being made by a qualified physician [20]. Adequate surgical care bundle adherence was defined to be present if a patient received at least six of the eight measures described in the bundle protocol. Secondary outcome measurements for possible adverse events were defined at 30 days postoperatively.

The following operative ICD breast surgery codes HAB00, HAB40, HAC10, HAC15, HAC20, HAC22, HAC30, HAD30, HAF00, HAC99 and ZZR70, in combination with axillary surgery codes PJA10, PJD42 and PJD52, were used for searching the hospital database system (Cognos Analytics^TM^). 

The inclusion process from all patients undergoing surgery during the study period was conducted in three consecutive stages: firstly, all NACT patients who underwent surgery were included, followed by all non-NACT patients who had breast operations combined with axillary clearances. Lastly, as the proportion of SLNB amongst the non-NACT group were predominantly larger than the axillary clearance, a random selection of those who had a breast operation with SLNB was conducted (Figure 1). 

### 2.3. Data Collection, Variables, and Predictors 

Clinically relevant information, including age, body mass index, smoking status, comorbidity (diabetes), types of surgery, surgical care bundle measures, SSI, microbiological cultures, chemo-radiotherapy, tumour biology, reoperations, postoperative thromboembolic events, seroma aspirations, length of stay and time to start of adjuvant treatments, were registered retrospectively. 

### 2.4. Statistical Analysis

The study cohort was divided into two groups depending on time periods: before and after SCB implementation. The impact of SCB on SSI was analysed using SPSS (Statistical Package for Social Science) version 28.0.1.0. Descriptive data for the variables were presented in absolute numbers and their percentages. Comparisons of proportions were calculated using the chi-squared analytical function. Risk factors for SSI were analysed using binary logistic regression using the Enter method including SCB implementation, age, BMI, smoking, diabetes, types of surgery, NACT and seroma aspirations. A statistical significance of *p* < 0.05 was used. Sample size calculation was not performed in this retrospective observational study.

## 3. Results

Out of all 3232 patients operated with non-reconstructive breast surgery, a total of 1132 patients were identified from the hospital registry following the inclusion criteria. Of those, 174 were excluded: 133 patients from the first 3 months of SCB introduction (introduction phase), 17 patients who underwent implant-based reconstructions, and 44 patients who underwent complex partial oncoplastic reconstructive surgery. In total, 958 patients were included in the study analyses (Figure 1). 

### 3.1. Cohort Characteristics

Patient characteristics were not statistically significant different before and after SCB implementations based on age, menopausal status or smoking status. All patients with diabetes had well-treated and stable disease with no differences between both groups. However, in the pre-SCB implementation period, there were statistically significant fewer BCSs (44.8% vs. 64.1%, *p* < 0.001), more mastectomies (55.2% vs. 35.9%, *p* < 0.001) and fewer patients receiving NACT (12.6% vs. 27.6%, *p* < 0.001). 

Tumour characteristics were not statistically significant different before and after SCB implementation based on tumour size, grade, and axillary status. However, in the pre-SCB implementation period, there were statistically significant fewer patients with Luminal-B (25.3% vs. 29.6%, *p* = 0.02) and HER2-luminal (7.9% vs. 12.9%, *p* = 0.02) breast cancer subtypes (Table 1).

In total, 10.4 % (100/958) of all included patients developed SSI. Microbiological cultures from wound or seroma aspirate were carried out in 53% (53/100) of patients with SSI but only 49.1% (26/53) showed clinically significant bacterial growth. Of these positive cultures with confirmed SSI, 84.6% (22/26) contained either staphylococcus or streptococcus bacteria.

### 3.2. Surgical Site Infections

After SCB implementation, there was a non-statistically significant absolute reduction in SSI rate of 2.9% (from 11.8% to 8.9%, *p* = 0.15). When comparing SSI rates following different types of surgery, there was a statistically significant absolute reduction of 9.0% (from 18.8% to 9.8%, *p* = 0.01) amongst patients who underwent BCS combined with SLNB, but a non-statistically significant reduction of 5.7% in SSI rate (from 8.8% to. 3.1%, *p* = 0.17) amongst patients who had BCS combined with axillary lymph nodal dissection (ALND). In comparisons of SSI rates amongst patients who underwent mastectomy, SCB implementation led to a non-statistically significant increase of 4.1% (from 6% to 10.1%, *p* = 0.24) in those who had SLNB, and a decrease of 3.3% (from 13.1% to 9.8%, *p* = 0.48) in those who underwent ALND. In further subgroup analyses of SSI rates in the NACT and non-NACT group, there were a non-statistically significant 4.4% reduction in SSI (from 10.9% to 6.5%, *p* = 0.28) and 2.1% (from 11.9% to 9.8%, *p* = 0.37), respectively, after SCB implementation (Table 2).

Of all SSI cases, 7.7% (74/958) occurred in the breast, 0.5% (5/958) in the axilla and 2.2% (*n* = 21/958) in both sites. These proportions were not statistically significantly different before or after SCB implementation (*p* = 0.32). 

### 3.3. Adherence of Surgical Care Bundle

An overall 22.0% of patients were already receiving SSI preventive measures on an ad hoc basis before the formal SCB implementation, with larger proportions in the NACT group (89.1%) compared with the non-NACT subgroup (12.4%) (Table 3). Following SCB implementation, the SCB adherence rate increased by 39.5% (*p* < 0.001). Amongst the subgroups, there was decrease in SCB adherence of 10.1% (*p* = 0.09) for NACT and an increase of 42.4% (*p* < 0.001) for non-NACT. The changes in SCB adherence rates for individual preventative measures are also summarized in Table 3. Specifically, prophylactic antibiotics were given within a similar timeframe preoperatively, before and after SCB implementation (26.4 min +/− SE 1.76 vs. 25.9 min +/− 1.33, *p* = 0.87, respectively).

### 3.4. Impact of Surgical Care Bundle

When adjusted for various patient and treatment factors, the implementation of a SCB led to an overall risk reduction of 37.0% in SSI (OR 0.63 95% CI 0.40–0.99, *p* = 0.04) (Table 4). Factors such as BMI, diabetes, mastectomy, and seroma aspirations were associated with statistically significant increases in risks of SSI. When SCB was unbundled into standalone preventative measures, neither uni- nor multivariate analyses on individual measures showed any statistically significant impacts on the SSI rate (data not shown).

### 3.5. Adverse Effects of Surgical Care Bundle

There was a small but non-statistically significant increase in thromboembolic events after stopping routine use of prophylactic anticoagulations (0.6% to 1.1%, *p* = 0.20). Likewise, there was a small but non-statistically significant increase in seroma aspirations after SCB implementation (23.6% to 25.6%, *p* = 0.46). Day surgery rates increased significantly (16.3% to 55.5%, *p* < 0.001). In contrast, there were non-statistically significant decreases both in re-operation rate due to bleeding (2.9% to 1.1%, *p* = 0.05) and in the proportion of patients who started their adjuvant chemoradiotherapy within 30 days (4.5% to 2.2%, *p* = 0.05), respectively (Table 5).

## 4. Discussion

The main aim of this study was to investigate if implementing SCB could lead to a reduction in SSI. This key finding indicated that SCB implementation was an independent factor associated with SSI risk reduction. The SCB adherence rate was 61% after SCB implementation with an increase of almost 40%. Risk factors associated with SSI were BMI, diabetes, mastectomy, and seroma aspirations. There was no adverse outcome caused by implementing a surgical care bundle to patient care pathways when they underwent non-reconstructive breast surgery.

However, there were limitations in our study. Sample size calculation was not performed, which limits the power of our study to detect any statistically significant differences in SSI outcome. Although there was a statistically significant SSI reduction in the BCS and ALND subgroups, it is important to point out that the case numbers in these subgroups were too small to draw any reliable conclusions from. There were also uncertainties inherent to the method of retrospective data collection. It is therefore difficult to reliably compare our results with other existing published data in a standardized way and therefore enable clinical application of SCB. Furthermore, comparison of outcomes in two different time periods can be difficult to interpret when factors other than SCB implementation can affect the results. For example, certain SCB components were meant to facilitate day surgery, which itself can be effective in SSI prevention. However, day surgery rates could also have been affected by other factors implemented for economic reasons. Additionally, surgical-site infection is defined by clinical observations as well as microbiological investigations. In our study, only half of SSI cases had microbiological investigations, which could cause inaccuracies in registering SSI events. However, not all SSIs have open wounds, and routine wound cultures may not be technically feasible. With only half of the wound cultures showing no or no significant bacterial growth, as well as staphylococcus and streptococcus being expectedly the commonest, it could be argued that microbiological cultures would not have added extra information to guide clinical treatments. That said, microbiological culture on open wounds or wound fluids should be considered more frequently as a measure of good practice. 

The primary endpoint with an adjusted analysis showed a decrease in the risk of SSI (OR 0.67, *p* = 0.04), while the overall absolute reduction in SSI rate did not reach statistical significance, which could be explained by various factors. Firstly, the fact that only 60% adhered to the SCB protocol despite formal implementation may have limited the care bundle’s impact in reducing SSI. Secondly, we found an unexpected proportion of both NACT and non-NACT patients already receiving SSI preventative measures sporadically before the formal SCB implementation. These pre-existing measures amongst patients could have limited the overall impact of SCB on SSI when implemented. It could also partially explain the paradoxical effect of NACT being associated with decreased SSI risks. Alternately, there was suggestion that NACT has an unexplained protective effect against SSI [21]. Thirdly, since there are currently no national or international guidelines on what would constitute an ideal and effective care bundle against SSI, it could mean that our SCB may not have been optimally constructed, and adjustments may be required. Lastly, normothermia during intraoperative and perioperative periods leads to lower SSI rate [22]. However, we were unable to reliably identify recordings of patient’s body temperatures in the study cohort, and it was possible that existent variations in body temperatures could explain the limited SCB impact on SSI reduction. Despite various confounding factors, we believed that SCB implementation increased the level of awareness amongst colleagues, thereby resulting in a significant reduction in SSI.

Unlike our study results, various standalone preventative measures had been shown to have effective impacts on SSI, but these reports were mostly in relation to reconstructive breast cancer surgery. Based on eleven randomized controlled trials (2867 participants) Gallagher et al. showed that preoperative antibiotics probably reduce the risk of SSI in breast surgery with a moderate grade of certainty [5]. In contrast, Liu et al. conducted a study on existing Cochrane reviews where data were extracted from 349 trials, totalling over 70,000 participants, which reported with high certainty evidence that preoperative antibiotic use reduces SSI [23]. Other standalone measures, such as antibacterial-coated sutures, have shown a significant effect in reducing SSI [24,25,26] in some, but not all, studies [27]. In addition, there were reports with variable degrees of certainty of reducing SSI by adopting wound irrigations and short-course postoperative antibiotics, as well as local anaesthetics with bactericidal effects [28,29]. Despite the above-mentioned reports on the impacts of individual measures against SSI, it is important to emphasise that effects in reducing SSI are best achieved through the synergistical actions of a standardised care bundle protocol instead. 

This study showed that mastectomy was associated with an increased SSI risk when compared with BCS procedures, which was in line with some recent studies [29]. However, based on subgroup analysis, SCB was not effective in reducing SSI after mastectomy in comparison with BCS (data not shown). This indicated that the impact of SCB may be selective and different strategies may be required to reduce SSI depending on types of breast cancer surgery. In addition, we also noted ALND was associated with a lower risk of SSI, although non-statistically significant. The cause for this paradoxical finding was unclear but potentially could be due to certain patient-related factors and further studies would be required.

Overall, there were no statistically significant adverse events in patient care following SCB implementation. Nevertheless, the reported small increase in seroma aspirations could potentially account for the low impact of SCB in reducing SSI as reported in our study. In addition, it was surprising to note that more patients had a delayed start of their adjuvant treatments following SCB implementation. The explanations for these treatment delays could be multifactorial, including patient and hospital resource factors. As this study was not aimed to investigate factors associated with treatment delays, we were therefore unable to accurately account for the potentially paradoxical relationship between SCB implementation and treatment delays.

## 5. Conclusions

In summary, our study demonstrated that implementation of an SCB could lead to a reduction in risk of SSI in non-reconstructive breast cancer surgery when adjusted for BMI, diabetes, types of surgery, NACT and seroma aspirations. Further prospective studies with an optimally constructed SCB protocol could be beneficial to further investigate the true effectiveness of SCB in improving patient safety.

## Figures and Tables

**Figure 1 cancers-15-00919-f001:**
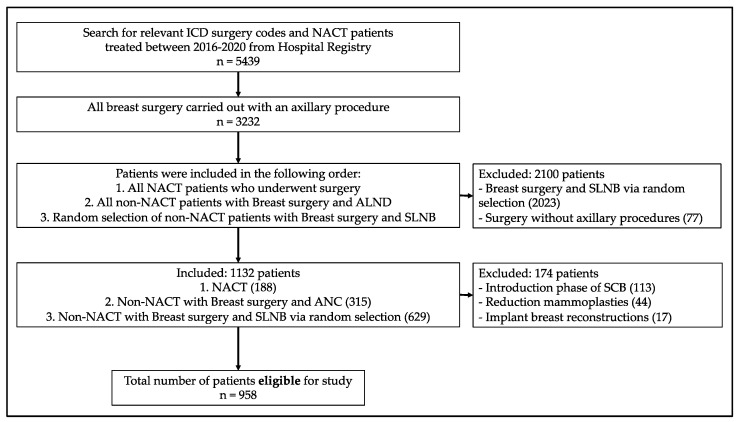
Flow chart for patient inclusions. Abbreviations: ALND: axillary lymph node dissection, NACT: neoadjuvant chemotherapy, RS: random selection, SLNB: sentinel lymph node biopsy, SCB: surgical care bundle.

**Table 1 cancers-15-00919-t001:** Cohort characteristics before and after implementation of the surgical care bundle.

		Before SCB *n* (%)	After SCB *n* (%)	*p*-Value
Proportions of patients		509 (53.1%)	449 (46.9%)	-
Mean age, years (range)		61.0 (29–94)	60.7 (21–94)	0.72
Menopausal status (cut-off age 50 years)	Pre	129 (25.3%)	104 (23.2%)	0.43
	Post	380 (74.7%)	345 (76.8%)	
Body mass index (kg/m^2^)	<30	413 (81.3%)	353 (79.0%)	0.37
	≥30	95 (18.7%)	94 (21.0%)	
Cigarette smoking	No	443 (87.0%)	397 (88.4%)	0.52
	Yes	66 (13.0%)	52 (11.6%)	
Diabetes	No	480 (94.3%)	419 (93.3%)	0.53
	Yes	29 (5.7%)	30 (6.7%)	
Breast surgery	BCS	228 (44.8%)	288 (64.1%)	<0.001
	Mastectomy	281 (55.2%)	161 (35.9%)	
Axillary surgery	SLNB	340 (66.8%)	303 (67.5%)	0.82
	ALND	169 (33.2%)	146 (32.5%)	
Neoadjuvant chemotherapy	No	445 (87.4%)	325 (72.4%)	<0.001
	Yes	64 (12.6%)	124 (27.6%)	
Tumour size (mm)	<20	184 (36.2%)	189 (42.3%)	0.15
	20–50	226 (44.4%)	177 (39.6%)	
	>50	99 (19.4%)	81 (18.1%)	
Histopathological grade	1	42 (8.8%)	44 (10.2%)	0.79
	2	280 (58.9%)	252 (58.2%)	
	3	153 (32.2%)	137 (31.6%)	
Tumour subtypes	Luminal-A	212 (41.7%)	155 (34.5%)	0.02
	Luminal-B	129 (25.3%)	133 (29.6%)	
	HER2-Luminal	40 (7.9%)	58 (12.9%)	
	HER2 non-luminal	37 (7.3%)	31 (6.9%)	
	TNBC	62 (12.2%)	55 (12.2%)	
	DCIS only	28 (5.5%)	14 (3.2%)	
	No cancer	1 (0.1%)	3 (0.7%)	
Axillary lymph node status	N0	341 (67.0%)	297 (66.1%)	0.78
	N+	168 (33.0%)	152 (33.9%)	

Abbreviations: ALND: axillary lymph node clearance, BCS: breast conservation surgery, DCIS: ductal carcinoma in situ, SCB: surgical care bundle, SLNB: sentinel lymph node biopsy, TNBC: triple-negative breast cancer.

**Table 2 cancers-15-00919-t002:** Surgical site infection rates before and after SCB implementations.

Patient Groups	Overall SSI Rate	SSI Rate before SCB Implementation	SSI Rate after SCB Implementation	Absolute Change SSI Rate	*p*-Value
Whole group (*n* = 958)					
Whole study cohort	10.4%	60/509 (11.8%)	40/449 (8.9%)	−2.9%	0.15
BCS + SLNB	13.5%	30/160 (18.8%)	22/224 (9.8%)	−9.0%	0.01
BCS + ALND	6.1%	6/68 (8.8%)	2/64 (3.1%)	−5.7%	0.17
Mastectomy + SLNB	7.3%	11/182 (6.0%)	8/79 (10.1%)	+4.1%	0.24
Mastectomy + ALND	11.6%	13/99 (13.1%)	8/82 (9.8%)	−3.3%	0.48
NACT (*n* = 188)					
Subgroup	8.0%	7/64 (10.9%)	8/124 (6.5%)	−4.4%	0.28
BCS + SLNB	0.0%	0/0 (0.0%)	0/31 (0.0%)	0.0%	-
BCS + ALND	7.3%	1/11 (9.1%)	2/30 (6.7%)	−2.4%	0.79
Mastectomy + SLNB	0.0%	0/3 (0.0%)	0/9 (0.0%)	0.0%	-
Mastectomy + ALND	11.6%	6/50 (12.0%)	6/54 (11.1%)	−0.9%	0.89
Non-NACT (*n* = 770)					
Subgroup	11.0%	53/445 (11.9%)	32/325 (9.8%)	−2.1%	0.37
BCS + SLNB	14.7%	30/160 (18.8%)	22/193 (11.4%)	−7.4%	0.05
BCS + ALND	5.5%	5/57 (8.8%)	0/34 (0.0%)	−8.8%	0.08
Mastectomy + SLNB	7.6%	11/179 (6.1%)	8/70 (11.4%)	+5.3%	0.16
Mastectomy + ALND	11.7%	7/49 (14.3%)	2/28 (7.1%)	−7.2%	0.35

Abbreviations: ALND: axillary lymph node clearance, BCS: breast conservation surgery, NACT: neoadjuvant chemotherapy, SCB: surgical care bundle, SLNB: sentinel lymph node biopsy, SSI: surgical site infection.

**Table 3 cancers-15-00919-t003:** Adherence rates before and after surgical care bundle implementation.

**Adherence of Individual** **Preventative SSI Measures**		**Before SCB** **Implementation** ***N* (%)**	**After SCB** **Implementation** ***N* (%)**	**Absolute Change** **(+/− %)**	***p*-Value**
1. Preoperative body wash with water and soap *	No	Missing data	Missing data	Not applicable	-
	Yes	Missing data	Missing data		
2. Antibiotic prophylaxis	No	300 (58.9%)	2 (0.4%)	+58.5%	<0.001
	Yes	209 (41.1%)	447 (99.6%)		
3. Wound irrigation	No	506 (99.4%)	228 (50.8%)	+48.6%	<0.001
	Yes	3 (0.6%)	221 (49.2%)		
4. Monofilament sutures with antibacterial coating	No	177 (34.8%)	4 (0.9%)	+33.9%	<0.001
	Yes	332 (65.2%)	445 (99.1%)		
5. Wound dressing (tape) *	No	Missing data	Missing data	Not applicable	-
	Yes	Missing data	Missing data		
6. Local anaesthetics	No	374 (73.5%)	18 (4.0%)	+69.5%	<0.001
	Yes	135 (26.5%)	431 (96.0%)		
7. Low molecular weight heparin	No	351 (69.0%)	425 (94.7%)	−25.7%	<0.001
	Yes	158 (31.0%)	24 (5.3%)		
8. Drains	No	244 (47.9%)	420 (93.5%)	−45.5%	<0.001
	Yes	265 (52.1%)	29 (6.5%)		
**SCB Adherence Based on Study Definition ****		**Before SCB** **Implementation** ***N* (%)**	**After SCB** **Implementation** ***N* (%)**	**Absolute Change** **(+/− %)**	***p*-Value**
For Whole group	No	397 (78.0%)	173 (38.5%)	+39.5%	<0.001
	Yes	112 (22.0%)	276 (61.5%)		
For NACT group	No	7 (10.9%)	26 (21.0%)	−10.1%	0.09
	Yes	57 (89.1%)	98 (79.0%)		
For Non-NACT group	No	390 (87.6%)	147 (45.2%)	+42.4%	<0.001
	Yes	55 (12.4%)	178 (54.8%)		

* Incomplete data capture for preoperative wash and surgical tapes usage as dressings. ** SCB adherence is defined as when patient received at least 6 of the 8 infection prevention measures.

**Table 4 cancers-15-00919-t004:** Uni- and multivariate analyses for different predictors of surgical site infections.

		Univariate Analysis			Multivariate Analysis		
		OR	95% CI	*p*-Value	OR	95% CI	*p*-Value
SCB implementation	Before	ref			ref		
	After	0.73	0.48–1.12	0.15	0.63	0.40–0.99	0.047
Age (years)		1.00	0.99–1.02	0.50	1.00	0.99–1.02	0.86
Body mass index		1.07	1.03–1.12	<0.001	1.07	1.03–1.11	0.001
Smoking:	No	ref			ref		
	Yes	1.41	0.79–2.50	0.24	1.57	0.86–2.84	0.14
Diabetes:	No	ref			ref		
	Yes	2.94	1.55–5.58	<0.001	2.17	1.10–4.30	0.03
Breast surgery:	BCS	ref			ref		
Mastectomy		0.76	0.50–1.15	0.19	1.74	1.07–2.84	0.03
Axillary surgery:	SLNB	ref			ref		
	ALND	0.86	0.55–1.35	0.52	0.80	0.47–1.37	0.42
NACT	No	ref			ref		
	Yes	0.70	0.39–1.24	0.22	0.90	0.44–1.83	0.77
Seroma aspirations:	No	ref			ref		
	Yes	1.52	0.97–2.38	0.07	2.06	1.22–3.47	0.007

Abbreviations: ALND: axillary node clearance, BCS: breast conservation surgery, NACT: neoadjuvant chemotherapy, SCB: surgical care bundle, SLNB: sentinel node biopsy.

**Table 5 cancers-15-00919-t005:** Secondary outcome measures in relation to implementation of surgical care bundle.

	Overall Rate *N* = 958	Before SCB *N* = 509 *n* (%)	After SCB *N* = 449 *n* (%)	Absolute Change +/− %	*p*-Value
Thromboembolic events	0.8%	3 (0.6%)	5 (1.1%)	+0.5%	0.20
Seroma aspiration	24.5%	120 (23.6%)	115 (25.6%)	+2%	0.46
Day surgery rate	34.7%	83 (16.3%)	249 (55.5%)	+39.2%	<0.001
Re-operation due to bleeding	2.1%	15 (2.9%)	5 (1.1%)	−1.8%	0.05
Adjuvant CRT started within 30 days	3.4%	23 (4.5%)	10 (2.2%)	−2.3%	0.05

Abbreviation: CRT: chemo- and radiotherapy.

## Data Availability

The data presented in this study are available on request from the corresponding author.

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
