# Peer review of "Impact of Surgical Care Bundle on Surgical Site Infection after Non-Reconstructive Breast Cancer Surgery: A Single-Centre Retrospective Comparative Cohort Study"

_cancers, 2023, doi:10.3390/cancers15030919_

Round 1
Reviewer 1 Report
This is a well written easy to follow observational study examining the implementation of a surgical care bundle on wound infection rates following breast cancer surgery without reconstruction.
This was a very labor intensive study. A sample size calculation might have reduced the amount of work that was required and came up with similar results with similar not very convincing evidence for change. Surprised that this was not done for the sampling of the patients that received breast conserving surgery with sentinel node biopsy. The intervention seemed reasonable with the myriad of potential items to include. No description of when antibiotics were given which is important if they are to be effective. The retrospective before and after study design is not a very strong methodology. Seemed to be other significant changes going on over the years that might have impacted the results such as the use of mastectomy, the use of neoadjuvant therapy, more day surgery and a change in the use of drains to facilitate day surgery. The subgroup analyses started to feel like a lot of data dredging, particularly with the unexpectedly high rates of wound infection in the breast conserving surgery and sentinel node biopsy group and large reduction in wound infection rates in this group with the surgical care bundle.
The references seem reasonable, however I did explore reference 25 and incidentally discovered that this paper by Prudencio did not mention the sutures they used. Thus the use of the Prudencio paper to support the statement “Other standalone measures like antibacterial coated suture have shown a significant effect in reducing SSI [24-26] in some but not all studies [27]" should be corrected. You have otherwise not referenced this paper anywhere else.
Given the not very strong study design and modest reduction in SSI in the overall analysis I find the conclusions a bit overstated: “Implementation of SCB led to a significant overall reduction of SSI.” I would have used a descriptor such as “modest” rather than significant and perhaps even “suggest” in the concluding line.
This paper is consistent with the literature that there are a lot of idiosyncrasies with wound infections in breast cancer surgery, such as higher infection rates than would be expected. I think the prior literature would already suggest that preoperative administration of prophylactic antibiotics may reduce infection rates among patients undergoing breast cancer surgery and that further studies are necessary to establish guidelines and protocols for clinical practice. This study, due to its design, provides little information to change that need.
Author Response
1:1 Comment: This was a very labor intensive study. A sample size calculation might have reduced the amount of work that was required and came up with similar results with similar not very convincing evidence for change. Surprised that this was not done for the sampling of the patients that received breast conserving surgery with sentinel node biopsy. The intervention seemed reasonable with the myriad of potential items to include. No description of when antibiotics were given which is important if they are to be effective. The retrospective before and after study design is not a very strong methodology. Seemed to be other significant changes going on over the years that might have impacted the results such as the use of mastectomy, the use of neoadjuvant therapy, more day surgery and a change in the use of drains to facilitate day surgery. The subgroup analyses started to feel like a lot of data dredging, particularly with the unexpectedly high rates of wound infection in the breast conserving surgery and sentinel node biopsy group and large reduction in wound infection rates in this group with the surgical care bundle.
Authors’ reply: We agree that a sample calculation could have been useful, however this was not performed. The instructions for prophylactic antibiotic defined that this should be given 2 hours up to 30 minutes before skin incision, this has now been clarified in the Methods section. We agree that a before/after design is not the best methodology, and that other factors changes as well during the time. However, we have tried to address this in our primary analysis by adjusting for these factors, including those mentioned by the reviewer. The decision to not use drains were included as a part of the SCB. We also would like to provide a counter explanation in relation to mentioned ‘data-dredging”. We acknowledge the problems in over-analysing subgroups in this study and have weighed against including these data. However, it is recognized that there is a wide variation between type of operations: ie BCS vs mastectomy, especially in the modern era of oncoplastic surgery. We believed that it is possible different SCB protocols will be required depending on techniques and types of surgery. For these reasons, we felt it is important to mention the difference in SCB impact outcome amongst the subgroups
1:2 The references seem reasonable, however I did explore reference 25 and incidentally discovered that this paper by Prudencio did not mention the sutures they used. Thus the use of the Prudencio paper to support the statement “Other standalone measures like antibacterial coated suture have shown a significant effect in reducing SSI [24-26] in some but not all studies [27]" should be corrected. You have otherwise not referenced this paper anywhere else.
Authors’ reply: We thank the reviewer for spotting this error, and have now removed the above mentioned reference and replaced it with another with the same ref nr: 25 (Uchino et al).
1:3 Comment: Given the not very strong study design and modest reduction in SSI in the overall analysis I find the conclusions a bit overstated: “Implementation of SCB led to a significant overall reduction of SSI.” I would have used a descriptor such as “modest” rather than significant and perhaps even “suggest” in the concluding line.
Authors’ reply: We agree with the reviewers’ point which was also raised by another reviewer. We have made changes accordingly in our conclusions and thank the reviewers for their advice.
Reviewer 2 Report
Dear authors,
Your study is addressing an important problem: the occurrence of SSI in patients with breast cancer surgery. As you have said in introduction, the incidence of this complications is still high and certain measures should be proposed in order to decrease the incidence. In this context your research is timely.
Regarding the results there are some aspects that should be clarified:
- The diabetes was compensated and well controlled in all patients? If not, are there statistical differences between groups?
- It was interesting to evaluate other parameters also (like the number of days of hospitalization or costs) in order to evaluate the impact of SCB
- In conclusion you should first mention that there was observed a decrease of SSI incidence but it was not statistically significant
Author Response
2:1 Comment: The diabetes was compensated and well controlled in all patients? If not, are there statistical differences between groups?
Authors’ reply: We can confirm that that all patients with diabetes had stable disease and if necessary optimized as part of their pre-operative work-up. We recognized that this is valid point and has included a statement of it in the result section.
2:2 Comment: It was interesting to evaluate other parameters also (like the number of days of hospitalization or costs) in order to evaluate the impact of SCB.
Authors’ reply: Thank you for this point which we had initially considered but opted against it as it may potentially confuse readers by mixing both clinical and costs data together. Therefore, we have not collected such information but thank the reviewer nevertheless for highlighting it.
2:3 Comment: In conclusion you should first mention that there was observed a decrease of SSI incidence but it was not statistically significant
Authors’ reply: The primary analysis of the study was the adjusted analysis, which was significant (p=0.04), and associated with quite high clinical difference (47% SSI risk reduction). However, we agree that with this type of study design, we have adjusted the conclusion in accordance with also the comment from Reviewer 1.
Reviewer 3 Report
The subject of this retrospective analysis is very important, the reduction of post-surgical wound infections in breast cancer surgery. The authors have performed a retrospective analysis comparing 2 cohorts, one before and one after implementation of a bundle of actions recommended by the WHO. The number of included patients seemed to be big enough to draw conclusions for the generation of hypotheses even if a calculation has not beeen done before.
The authors found a nonsignificant reduction of SSI (surgical site infections) in the primary analysis with a significant reduction in patients with BCT and SLNB, nonsignificant reductions in most other groups and and a nonsignificant increase in patients with mastectomy and SLNB. These heterogenous results lead to the question if the case number really was appropriate. Although the decrease in SSI was significant in the multivariate analysis it remains questionable how solid these results are regarding the huge number of influencing factors in the incidence of SSI. Although the authors are discussing many of these factors elegenatly, what is missing in the discussion is the statement that maybe the case number was simply too small for a real conclusion. The authors' conclusion is reflecting the need for a prospective trial (that should be easily recruited and yield a fast read-out), only the need for a very detailed sample size calculation incorporating thze estimated effect sizes of influencing factors and factoring them in should be added.
Author Response
3:1 Comment: The authors found a nonsignificant reduction of SSI (surgical site infections) in the primary analysis with a significant reduction in patients with BCT and SLNB, nonsignificant reductions in most other groups and and a nonsignificant increase in patients with mastectomy and SLNB. These heterogenous results lead to the question if the case number really was appropriate. Although the decrease in SSI was significant in the multivariate analysis it remains questionable how solid these results are regarding the huge number of influencing factors in the incidence of SSI. Although the authors are discussing many of these factors elegenatly, what is missing in the discussion is the statement that maybe the case number was simply too small for a real conclusion.
Authors’ reply: The primary analysis was the multivariate analysis, adjusting for the confounding factors that changed over the study period, and that gave a statistically and clinical meaningful reduction in SSI risk. However, we agree with the missing discussion concerning included number of patients that was missing in the initial discussion and we thank the reviewer for highlighting this. We have now added a sentence in the discussion addressing this.
3:2 Comment: The authors' conclusion is reflecting the need for a prospective trial (that should be easily recruited and yield a fast read-out), only the need for a very detailed sample size calculation incorporating thze estimated effect sizes of influencing factors and factoring them in should be added.
Authors’ reply: We would like to thank the reviewer for highlighting the above suggestions for an onward study to delineate further the true impact of SCB on SSI with more specific study design approaches. It would be beyond the scope of this paper for a detailed discussion but we acknowledged the importance of the reviewer’s point nevertheless.